# Reduction in Nuclear Size by DHRS7 in Prostate Cancer Cells and by Estradiol Propionate in DHRS7-Depleted Cells

**DOI:** 10.3390/cells13010057

**Published:** 2023-12-27

**Authors:** Andrea Rizzotto, Sylvain Tollis, Nhan T. Pham, Yijing Zheng, Maria Alba Abad, Jan Wildenhain, A. Arockia Jeyaprakash, Manfred Auer, Mike Tyers, Eric C. Schirmer

**Affiliations:** 1The Institute of Cell Biology, University of Edinburgh, Edinburgh EH9 3BF, UK; andrea.rizzotto88@gmail.com (A.R.); jeyaprakash.arulanandam@ed.ac.uk (A.A.J.); 2Institute of Biomedicine, University of Eastern Finland, 70210 Kuopio, Finland; sylvain.tollis@uef.fi; 3The Institute of Quantitative Biology, Biochemistry and Biotechnology, University of Edinburgh, Edinburgh EH9 3BF, UK; nhan.pham@ed.ac.uk (N.T.P.); yzheng10@gmail.com (Y.Z.); jan.wildenhain@gmail.com (J.W.); manfred.auer@ed.ac.uk (M.A.); 4Wellcome Centre for Cell Biology, University of Edinburgh, Edinburgh EH9 3BF, UK; mabadfe@exseed.ed.ac.uk; 5Gene Center and Department of Biochemistry, LMU-München, 81377 Munich, Germany; 6Xenobe Research Institute, P.O. Box 3052, San Diego, CA 92163-1052, USA; 7Program in Molecular Medicine, The Hospital for Sick Children, Toronto, ON M5G 0A4, Canada; mike.tyers@sickkids.ca; 8Department of Molecular Genetics, University of Toronto, Toronto, ON M5S 1A8, Canada

**Keywords:** nuclear size regulation, cancer, metastasis, NET50, androgen-insensitive

## Abstract

Increased nuclear size correlates with lower survival rates and higher grades for prostate cancer. The short-chain dehydrogenase/reductase (SDR) family member DHRS7 was suggested as a biomarker for use in prostate cancer grading because it is largely lost in higher-grade tumors. Here, we found that reduction in DHRS7 from the LNCaP prostate cancer cell line with normally high levels of DHRS7 increases nuclear size, potentially explaining the nuclear size increase observed in higher-grade prostate tumors where it is lost. An exogenous expression of DHRS7 in the PC3 prostate cancer cell line with normally low DHRS7 levels correspondingly decreases nuclear size. We separately tested 80 compounds from the Microsource Spectrum library for their ability to restore normal smaller nuclear size to PC3 cells, finding that estradiol propionate had the same effect as the re-expression of DHRS7 in PC3 cells. However, the drug had no effect on LNCaP cells or PC3 cells re-expressing DHRS7. We speculate that separately reported beneficial effects of estrogens in androgen-independent prostate cancer may only occur with the loss of DHRS7/ increased nuclear size, and thus propose DHRS7 levels and nuclear size as potential biomarkers for the likely effectiveness of estrogen-based treatments.

## 1. Introduction

Nuclear size and shape changes have been used in cancer diagnostics for >150 years [1] and for many specific cancer types [2], among which is prostate cancer (PCa), where an increased nuclear size generally correlates with a higher Gleason score, metastasis, and poor survival [3,4,5,6,7]. However, it remained unclear whether nuclear size changes in the various tumor types are an indirect consequence or a driver of the tumor grade/increased metastasis. We recently found that the correction of cancer-associated nuclear size changes in different cancer-type cell lines can reduce cell migration and invasion [8], suggesting that reversing nuclear size changes might benefit anti-cancer therapies. As nuclear size changes are characteristic for each cancer type, we predicted that nuclear envelope transmembrane proteins (NETs), which have highly tissue-specific expression patterns, might mediate these changes. We therefore became interested in DHRS7, a protein that largely localizes in the nuclear envelope [9], is expressed the highest in healthy prostate cells [10,11] and has decreased expression in higher-grade PCa [12].

DHRS7 (also called NET50, retSDR4, and SDR34C1) is an orphan member of the short-chain dehydrogenase/ reductase (SDR) family involved in the metabolism of steroids and retinoids [13,14]. DHRS7 reduces endogenous steroids like estrone, cortisone, and 4-androstene-3,17-dione, and exogenous molecules presenting a carbonyl group like 1,2-naphtoquinolone [12,13,14]. DHRS7 was thus suggested to sustain de novo androgen synthesis and/or lead to a reactivation of the androgen receptor [15]; however, the DHRS7-catalyzed conversion of 5α-dihydrotestosterone to 3α-androstanediol was alternatively suggested to reduce androgen receptor transcriptional activity [16]. These results, although conflicting, both argue that the sterol dehydrogenase function of DHRS7 might impact on the androgen-sensitive state of PCa cells.

DHRS7 was amongst the genes altered in expression in PCa according to several studies [17,18,19,20,21]. DHRS7 was also strongly altered in an a LNCaP hollow fiber model in mice at the castration-recurrent stage [15], and highly downregulated in metastatic PCa compared to non-recurrent primary PCa [22]. More focused investigations showed that DHRS7 levels strongly dropped in higher-grade PCa [12], DHRS7 expression was abolished in metastatic PCa and that survival was increased for patients with higher DHRS7 levels [23]. Accordingly, DHRS7 is expressed in the PCa LNCaP androgen-dependent cell line model but not in the PC3 androgen-independent cell line model [12]. Furthermore, the directed specific loss of DHRS7 in LNCaP correlated with increased proliferation and cell migration and decreased cell adhesion [12]. Taken together, these results argue for the mechanistic role of DHRS7 in PCa evolution toward metastasis. Nevertheless, how DHRS7 is involved and if it is directly involved in the metastatic evolution of PCa is still unclear.

Here, we found that the knockdown of DHRS7 in LNCaP cells increased nuclear size, while nuclear size was decreased with an overexpression of DHRS7 in PC3 cells. Due to the androgen independence of castration-recurrent PCa, estradiol was proposed as a treatment thanks to its indirect negative feedback in androgen production [24,25]. We found that the treatment of PC3 cells lacking DHRS7 with estradiol propionate also reduced nuclear size, but the effect of estradiol propionate was lost in presence of DHRS7. Mutational analysis indicated that the nuclear size–regulatory function of DHRS7 is dependent on its enzymatic activity. Together with recent findings that nuclear size correction can reduce cell migration/ invasion aspects of metastasis [8], these findings strengthen the case for targeting the DHRS7 nexus in treating PCa.

## 2. Materials and Methods

### 2.1. Cell Culture and Transfection

PC3 (ATCC CRL-1435), PC3-H2B-RFP (stable plasmid integration in PC3), and HT1080 fibrosarcoma (ATCC CCL-121) cell lines were grown in DMEM, 10% fetal bovine serum (FBS). LNCaP (ATCC CRL-1740) cells were grown in RPMI 1640, 10% FBS. Media were supplemented with 100 U/ml penicillin and 100 mg/ml streptomycin, except prior to drug screening. For experiments with hormones, FBS was charcoal-stripped. Cells were plated onto 13 mm coverslips in 35 mm dishes and transfected at 40% confluency with 1.6 μg of plasmid using Lipofectamine 2000 (Invitrogen, Waltham, MA, USA).

### 2.2. Plasmids and Construction of Stable Cell Lines

Human DHRS7 was expressed as a GFP fusion using the pEGFP-N2 vector (Clontech-Takara Bio, Mountain View, CA, USA) generated previously [9]. To generate PC3-H2B-mRFP, the H2B-mRFP plasmid carrying the blasticidin resistance gene was linearized and co-transfected with plasmid pToI2 encoding for transposase using the JetPrime (PolyPlus, Loos, France) transfection reagent. Then, 24 h post transfection, the medium was replaced with a medium containing 8 μg/ml blasticidin (Thermo Fisher Scientific, Waltham, MA, USA), which was maintained for ~1 week to select stably integrated transfectants. After amplification, the cells were FACS-sorted for the brighter mRFP population that retained normal morphology. Subsequently, the passages of cells were alternated between selection and no selection; however, the cells used in experiments were always following a passage without blasticidin and the screening was performed in the absence of blasticidin.

### 2.3. DHRS7 Knockdown

For the knockdowns, 1.5 × 10^6^ cells were seeded the day before transfection to attach to 6-well plates. The following morning, 25 pmol of Dharmacon Smart Pool *DHRS7* L-009573-00-0005 siRNA (GAAUGGGAGCUGACUGAUA, CAGCAUGGCCAAUGAUUUG, GCUAAUAGAGCUUAACUAC, and GGAUGCAGACUCUUCUUAU) was mixed with 4 μL of JetPrime (PolyPlus, Loos, France) transfection reagent in 200 μL of the JetPrime buffer and added to the cells at ~10 nM. Cells were allowed 72 h before analysis of protein levels via Western blot or fixation for immunofluorescence. For DHRS7 knockdown experiments in the presence of estradiol propionate, the drug was added at the time of transfection with DHRS7 siRNAs and maintained for the duration of the experiment.

### 2.4. Antibodies and Western Blotting

Cell lysates were resolved by 12% SDS-PAGE. Proteins were transferred onto nitrocellulose membrane using a Pierce Power station (Thermo-Fisher Scientific, Waltham, MA, USA). Membranes were blocked in 4% non-fat milk in PBS+0.1% Tween-20 for 1 h and incubated in the same buffer with rabbit anti-DHRS7 (Abcam, Cambridge, UK; ab156021; 1:1000) and anti-tubulin (Sigma, St Louis, MO, USA; T6074; 1:2000) antibodies for 1 h at RT. Antibody signals were scanned on a LI-COR Odyssey imager (LI-COR Biosciences, Lincoln, NE, USA) using InfraRed Dye-conjugated secondary antibodies. Scans were opened in Fiji and standard measurements taken for an identically sized region of interest applied to all bands and backgrounds. Median values were extracted and the background from the same lane subtracted. Corrected DHRS7 values were normalized to tubulin loading and percentage of the LNCaP control DHRS7 signal calculated for the other lanes.

### 2.5. Digitonin-Permeabilized Cell Assay

To determine membrane topology, fixed DHRS7-GFP-transfected cells (tag at the C-terminus) were either permeabilized with 150 µg/ml digitonin in PBS or 0.2% Triton X-100 for 7 min on a pre-cooled metal block sitting in ice. Detergents were removed by washing in PBS and the cells processed for immunofluorescence with polyclonal rabbit anti-GFP antibodies (made in-house) at 1:100 dilution that were visualized using Alexa-fluor 568 secondary antibodies. If the GFP antibody signal was present in the Triton-treated samples but not the digitonin-treated samples, then the C-terminus was in the lumen. If the GFP antibody signal was present in both, then the C-terminus faced the cytoplasm/nucleoplasm. In this latter case, if the nuclear rim signal was comparatively weaker than the ER signal for the digitonin compared to Triton samples, then the protein was in the inner nuclear membrane.

### 2.6. Fluorescence Microscopy and Image Analysis for Nuclear Volume Measurements

Plasmid-transfected cells were fixed with 4% PFA in PBS for 7 min at 48 h after transfection, while siRNA-transfected cells were fixed at 72 h. Where antibody staining was required, coverslips were permeabilized with 0.2% Triton X-100 for 10 min and blocked in 4% BSA in PBS for 20 min followed by incubation with relevant antibodies (DHRS7 Abcam, Cambridge, UK at 1:1000 dilution or rabbit lamin A antibodies [26] at 1:50 dilution for 1 h at RT) and subsequent staining with 4’,6-diamidino-2-phenylindole (DAPI) at 1:2000 in PBS to visualize DNA. Coverslips were mounted with VectaShield (VectaLabs, Castle Hill, Australia). Z-stack images at 0.2 µm steps were obtained using a Nikon TE-2000 microscope equipped with a 1.45 NA 100× objective, Sedat quad filter set, PIFOC Z-axis focus drive (Physik Instruments, Karlsruhe, Germany), and CoolSnapHQ High Speed Monochrome CCD camera (Photometrics, Tucson, AZ, USA) run using Metamorph version 7.0 software. Image deconvolution was performed with AutoQantX (Media Cybernetics, Rockville, MD, USA) and 3D-volume reconstructions generated in Image 3D-Pro (Media Cybernetics), applying standard settings. Statistics were performed with Prism 6 (Graphpad) using the unpaired *t*-test with the Welch correction. As 3D measurements are both more accurate and take much longer, statistical analysis was performed at various times during image collection so that some experiments had more cells analyzed than others, though generally, a minimum of 30 cells was analyzed per condition.

### 2.7. Library Screening

Plate 6 from the MicroSource Spectrum compound library (Discovery Systems, Inc., Mount Prospect, IL, USA) contained 80 different compounds at 1 mM in DMSO and 16 in-plate DMSO controls (see Appendix A). For each biological replicate performed on different days, 5000 PC3-H2B-RFP cells were plated on each well of Greiner Screenstar 96-well glass-bottom imaging plates and left overnight in 99 µL medium for adhesion and growth prior to compound treatment. Compounds were added by a Biomek (Beckman Coulter, Indianapolis, IN, USA) automated liquid handler robotic platform pipetting 1 µL from each well of the library plate into the cell plate (final concentration 10 µM), and cells were incubated for 36 h before fixation in 3.7% formaldehyde (EMD FX0410-5 formaldehyde solution) for 15 min at RT, washed with PBS, then incubated 30 min in 50 μL HCS CellMask Deep Red 0.5X (Molecular Probes, Leiden, Netherlands) for cytosol staining.

Cells were imaged using an Opera HCS imaging system (PerkinElmer), equipped with a 20X air objective (LUCPLFLN, NA = 0.45). H2B-mRFP and CellMask Deep Red fluorochromes were excited sequentially by 561 and 640 nm lasers, respectively, and emitted fluorescence acquired by two Peltier-cooled CCD cameras with bandpass detection filters at 600 (600/40) and 690 (690/70) nm, respectively, and the respective exposure times of 320 and 200 ms. For each well, 20 fields of view (FOVs) were acquired at the same positions for all wells and plates, allowing for the quantification of nuclear and cellular size for 150–2000 cells for each compound per replicate, except for estradiol propionate and fenbendazole that were slightly lower due to high cell death (see Appendix A). We note that compounds that strongly reduced nuclear size also reduced the number of cells scored, indicating an effect on viability. An adapted Acapella^®^ (PerkinElmer) software script was used to automatically mask the cytoplasm and nucleus of individual cells, as performed previously [8]. The script parameters are provided in Appendix A. Cell and nuclear sizes were defined as the areas (in pixels) of the masked regions in the focal plane, based on the CellMask Deep Red and H2B-mRFP signals, respectively.

### 2.8. Generation of Catalytic Site Mutations

Site-directed mutagenesis was performed on the pEGFP-N2-DHRS7 vector according to the manufacturer’s recommendations using the QuikChange II Site-Directed Mutagenesis Kit (Agilent), with primers designed using the Agilent on-line tool (https://www.agilent.com/store/primerDesignProgram.jsp (accessed on 1 July 2017)).

### 2.9. Bioinformatics

The homology model of DHRS7 was obtained with the Phyre2 web server, http://www.sbg.bio.ic.ac.uk/phyre2/ (accessed on 1 April 2017) using HSD1, human 11-beta-hydroxysteroid dehydrogenase 1—PDB: 2ILT [27] for the model shown in Figure 5. The expression levels of DHRS7 in different tissues were compared using data downloaded from the U1331 dataset of http://biogps.gnf.org [11] in January 2020 using probeset 210788_s_at. The median expression value over all 84 tissues analyzed was determined and the fold-expression over this value calculated for the individual tissues shown in Figure 1A.

## 3. Results

### 3.1. DHRS7 Is in the Nuclear Envelope and Contributes to Nuclear Size Regulation in PCa Cells

DHRS7 is expressed in a subset of tissues and most highly in prostate, where it is expressed 34-fold higher than the median value across 84 human tissues sampled according to BioGPS (Figure 1A) [10,11]. The exogenously expressed protein was previously shown to target the inner nuclear membrane in the HT1080 fibroblast cancer cell line [9]; however, functional DHRS7 has been assumed to localize in the endoplasmic reticulum (ER) in PCa models.

To gain an insight on the involvement of DHRS7 in PCa, we used two PCa model cell lines: LNCaP that was established from a metastatic lesion in a lymph node and is androgen-dependent [25,28,29,30,31,32]; and PC3 that is derived from a PCa lumbar metastatic lesion and is androgen-independent [33]. We found that DHRS7 was highly expressed in LNCaP cells (Figure 1B), where it is localized mostly in the nuclear envelope and much less in the ER (Figure 1C), but the levels dropped close to the background in PC3 cells (Figure 1B), in agreement with previous reports [28,29,33]. DHRS7 localization suggested that its function(s) at the nuclear envelope could be relevant in prostate.

DHRS7 is predicted by TMHMM v.2.0 to have a single N-terminal transmembrane helix (Figure 1D), but the topology of the protein is still debated. It was originally expected that the catalytic domain would face the ER lumen [13], but its activity is not stimulated by ER luminal H6PDH-mediated NADPH generation, suggesting a different orientation. Thus, a cytoplasmic orientation of the catalytic moiety was proposed based on similarity with other SDR family members [16]. To test if this was the orientation in the nuclear envelope, we performed a digitonin protein topology assay [34]. This revealed that DHRS7 presents its C-terminus (containing the catalytic domain) to the cytoplasm or nucleoplasm depending on whether it is in the ER/ outer nuclear membrane or the inner nuclear membrane (Figure 1E). This is in line with the topology experimentally determined for related dehydrogenase/reductase family members 11β-HDS and 17β-HDS that are involved in the control of active and inactive androgens and estrogens [16]. Because DHRS7 was previously confirmed in the inner nuclear membrane of HT1080 cells (derived from a fibrosarcoma [9]), our result indicates that its catalytic domain could face the nucleoplasm and therefore directly contribute to nuclear functions.

As DHRS7 is highly down-regulated during PCa progression [12] when nuclear size increases [6], we sought to check the effect of DHRS7 knockdown on nuclear size in LNCaP cells where it is normally highly expressed. siRNA-mediated knockdown to ≤20% was confirmed via Western blot (Figure 2A) and nuclear volume was measured from 3D reconstructions. Nuclear volume increased upon loss of DHRS7 (Figure 2B) to an extent similar to the increase that correlates with the switch from low-grade to later-stage prostate cancer.

We next exogenously expressed DHRS7 in PC3 cells that normally have very low DHRS7 levels, and in HT1080 cells which also have very low endogenous DHRS7 levels (Figure 2C). DHRS7 expression reduced nuclear size in both cell lines (Figure 2D). Hence, DHRS7 down-regulation may be sufficient to explain most of the nuclear size increase that occurs as the tumor grade increases in PCa. We predict that these nuclear size changes drive the reported increased cell migration of LNCaP cells knocked down for DHRS7 [12] as we previously showed that correcting the nuclear size defect in several cancer cell lines reduced cell migration/invasion [8].

### 3.2. A Partial Screen for Compounds Reverting Metastatic Nuclear Size Changes in PCa Identifies Estradiol Propionate

As an alternative to trying to express DHRS7 to reduce nuclear size in PCa, we set up a screen to identify compounds that could similarly reduce nuclear size, hoping that such compounds might also shed light on how DHRS7 reduces nuclear size. We tested 80 compounds from the MicroSource Spectrum chemical library for their ability to reduce nuclear size in PC3 cells. The cells were stably transfected with H2B-RFP so that the nuclear area could be measured from the RFP signal using fluorescence microscopy (Figure 3A). Cells were treated for 36 h with the compounds at 10 μM (Appendix A), then fixed and stained with CellMask Deep Red dye to also determine the total cell area and imaged using a PerkinElmer OPERA^TM^ high-throughput confocal microscope plate reader. Imaging data (150–2000 cells per condition) were analyzed using the built-in Acapella scripting environment (PerkinElmer), as previously done in [8].

Only 6 of the 80 compounds reproducibly reduced the cell population-averaged nuclear size more than 20% compared to in-plate DMSO controls (Figure 3B): fenbendazole (well-B3), pimozide (well-C10), sulconazole nitrate (well-D11), suloctidil (well-E6), estradiol propionate (EP; well-F4), and econazole nitrate (well-F7). Sulconazole and econazole are imidazole derivatives with antifungal activity. Fenbendazole is a benzimidazole antihelmintic compound. Notably, other antihelmintic imidazole derivatives were recently shown to reduce PC3 nuclear size [8]. EP is a direct derivative of estradiol bearing a carbonyl group instead of a single oxygen at C17β (Figure 3C). Other estradiol derivatives tested in our screen (Figure 3B: two dots directly to the right of and above estradiol propionate—estradiol benzoate well-F5, estradiol acetate well-F6) did not affect nuclear size as can be further observed by comparing EP and estradiol benzoate images (Figure 3D), consistent with different derivatives having potentially drastically different effects on nuclear size. While estradiol is used therapeutically for castration-recurrent PCa due to its negative feedback on androgen production [24,25], EP was never tested. Given that EP reverses the direction of cancer-associated nuclear size changes in PC3, and that DHRS7 expression status recapitulates nuclear size changes in progressing PCa, we sought to investigate whether EP acts in a DHRS7-dependent way to regulate nuclear size.

### 3.3. EP Rectifies Nuclear Size Changes in PCa Cells Only When DHRS7 Is Reduced

We tested for chemical–genetic interactions by measuring the nuclear size changes in EP-treated PC3 and LNCaP cells +/− targeting DHRS7 levels. EP, both at 3 μM and 10 μM concentrations, had no effect on nuclear volume in DHRS7-positive (wild type) LNCaP cells (Figure 4A). However, when DHRS7 was knocked down, the LNCaP cells became sensitive to EP nuclear size effects in a dose-dependent manner (Figure 4B). This quantitatively mirrored EP-driven nuclear size reduction in wild-type, naturally DHRS7-reduced PC3 cells (Figure 4C). Hence, low DHRS7 is sufficient to elicit EP-driven reduction in nuclear size in two PCa cell lines. Strikingly, DHRS7 loss may also be necessary since restoration of DHRS7 expression in PC3 cells abrogated the nuclear size effects of EP (Figure 4D). The change in nuclear size was ~20–30% for both DHRS7 loss and EP treatment, suggesting that both regulate nuclear size via common pathways.

### 3.4. Predicted Catalytically Dead DHRS7 Mutant Loses Its Nuclear Size Altering Function

We next sought to determine if the catalytic function of DHRS7 is involved in its nuclear size–regulatory function. As the catalytic/cofactor-binding sites of other members of the SDR family, but not DHRS7, have been experimentally determined, we relied on sequence–structure relationships. A crystal structure of SDR family member 17β-hydroxysteroid-dehydrogenase type 1 (17β-HSD-1) revealed R37 stabilizing the 2’-phosphate of cofactor NADP^+^ [35]. Bioinformatics analysis across the SDR family identified the equivalent arginine in 11β-HSD-B1 as R66 and for DHRS7 as R82 [36] (Figure 5A). Using an 11β-HSD-B1-based [37] homology-modeled structure of DHRS7 (Figure 5B), we confirmed that R82, together with R83, forms the NADP^+^ binding pocket (the alpha fold model of DHRS7 yielded the same results) and predicted which mutations of R82 would critically disrupt the NADP^+^ binding pocket. This predicted that an R82E mutation in DHRS7 would make it non-functional, consistent with experiments on 17β-HSD-3 where the R-to-E mutation was catalytically dead [38].

Thus, a DHRS7-R82E mutant was generated to test if the mutant still has the nuclear size altering function. Both mutant and wild type were fused to EGFP to identify transfected cells in microscopy experiments. While nuclear size reduction was observed in PC3 cells expressing wild-type EGFP-DHRS7, nuclear size was unaffected upon the expression of the EGFP-DHRS7-R82E mutant protein (Figure 5C). Whereas treatment with EP had no effect on PC3 cells expressing wild-type DHRS7 (Figure 4D), EP was able to reduce nuclear size when the R82E mutation was expressed (Figure 5D). Hence, the R82E mutation in DHRS7 abrogates DHRS7 effects on nuclear size and the EP-DHRS7 chemical–genetic interaction.

## 4. Discussion

Metastatic tumor treatments often target microtubules to inhibit both cell migration and division. While effective, this is extremely toxic to healthy cells; therefore, patients could benefit from drugs that have more tissue-specific effects. Nuclear size changes occur when cancers progress to being higher-grade, and more metastatic tumors tend to be highly characteristic for different tissue/tumor types [2,3,4,5,6,7,39]. These size changes likely facilitate metastatic spread by altering connections between the nuclear envelope and the cytoskeleton that can have a huge impact on cell migration [40]. Consistent with these points, we previously showed that drugs rectifying nuclear size defects in three cell lines modeling distinct tissue/tumor types, including PC3 cells representing PCa, are mostly tissue/tumor-type-specific and reduce cell migration and invasion [8]. Therefore, drugs targeting tissue/tumor-type-specific nuclear size changes might provide less systemically toxic therapies [41], and more so if they specifically target cancer cells within the target tissue.

In this work, we potentially identified such a drug: EP only alters nuclear size when DHRS7 is absent, which is characteristic of higher-grade, more metastatic PCa [12,22,23]. As DHRS7 is highly expressed in normal prostate, EP would not be expected to have negative effects—at least with respect to nuclear size changes—on the rest of the tissue. Rather, EP would be expected to specifically inhibit migration of the presumably more metastatic cells that have altered nucleo-cytoskeletal connections due to the size increase, thus biasing clonal selection toward less aggressive cells. Importantly, we identified a(n) (epi)genetic requirement—the loss of DHRS7—to elicit the nuclear size response to EP; hence DHRS7 could potentially be a predictor of success in estrogen PCa therapies, though this needs to be directly tested.

The high amount of DHRS7 in the nuclear envelope and its effect on nuclear size suggest that it functions predominantly in the nucleus, despite its multiple localizations. Many proteins have multiple subcellular locations and such proteins in the nuclear envelope often assume distinct functions. For example, lamin B receptor has a C14-sterol reductase function, yet it also binds heterochromatin proteins [42,43,44]. Therefore, we anticipated that DHRS7’s nuclear membrane localization, which being dependent on lamin A indicates inner nuclear membrane localization [9], might suggest a non-enzymatic mechanism of DHRS7′s role in nuclear size regulation. However, our finding that its dehydrogenase/reductase function appears to be required for this role would seem to counter this prediction.

Though how DHRS7 and EP can alter nuclear size remains unclear, it is possible that they function in the same pathway, keeping nuclear size to a “minimum”; so, if DHRS7 function saturates the pathway, then EP cannot have a further effect. For example, DHRS7 could contribute to producing a steroid hormone that cannot be made by other SDR family members. If this hormone promotes the expression of genes that cause the nuclear size change or maintains homeostasis that keeps nuclei small, the loss of DHRS7 would yield the nuclear size increase. EP could potentially fulfil the same function of that steroid hormone in regulating gene expression. Alternatively, DHRS7 catalytic activity could convert EP to a non-functional form. These are among many possibilities that are consistent with the EP functioning in the absence of DHRS7 or with its apparent catalytic inactivation.

## 5. Conclusions

Even without knowing the specific mechanism, our finding that EP can reverse nuclear size changes that typically occur in higher-grade PCa may potentially be exploited therapeutically. Estrogens are used to treat breast cancer, possibly contributing multiple benefits as they also yield a reduction in nuclear size in the MCF7-10F breast cancer cell line [45,46]. Estradiol is also effective in orchiectomized mouse xenograft models of PCa [47] and has been used in some treatments of castration-recurrent PCa (CRPC) based on its seemingly downregulating androgen production via negative feedback control [24,25]. In cell culture experiments, estradiol and diethylstilbestrol killed PCa cells with different experiments, suggesting that this additional effect was due to the activation of different microtubule-associated proteins (MAPs) that generate signaling cascades for the production of either reactive oxygen species (ROS) or induction of caspases, leading to necrosis or apoptosis, respectively [48]. However, there are arguments also against the use of estradiol because of variability in its effect on different patients [24]. As estradiol was also reported as one of DHRS7’s substrates [49], it is possible that even small amounts of DHRS7 are sufficient to convert estradiol and block its function. In this case, DHRS7 levels could be used as a biomarker for the likelihood of estrogen therapies being effective, though this needs to be directly tested in patients. EP is a direct estradiol derivative that has not been found in nature and may have anti-PCa benefits over estradiol. This possibility is strengthened by our finding that two other estradiol derivatives on the plate had no effect on nuclear size. While these implications of our findings need to be directly tested biochemically, in primary patient cells, and in animal models, our finding that EP can reverse the direction of nuclear size changes that typically occur in higher-grade and more metastatic PCa raises both the possibility that EP may prove better than estradiol and that screening for the absence of DHRS7 prior to engaging estradiol treatment may improve the outcomes in treating androgen-independent PCa.

## Figures and Tables

**Figure 1 cells-13-00057-f001:**
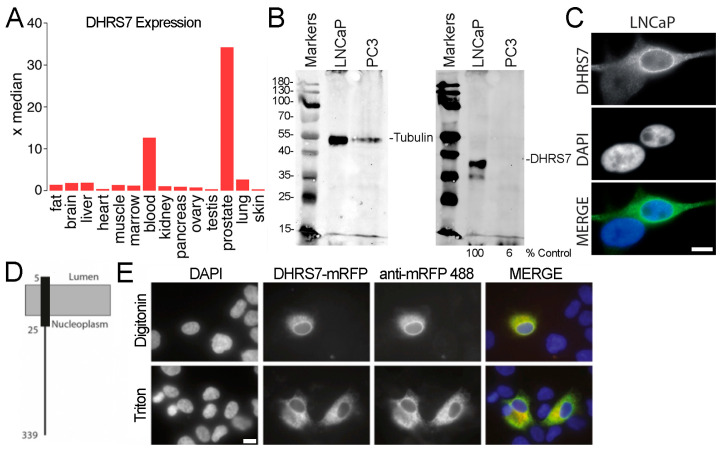
DHRS7 expression, subcellular localization, and orientation. (**A**) Plotted as x-fold over the median value across 84 human tissues, expression data extracted from BioGPS reveals DHRS7 to be highly preferentially expressed in normal prostate (14 extracted tissues shown). (**B**) Levels of DHRS7 protein in LNCaP and PC3 cells determined by Western blot. Numbers on the left indicate molecular weights. The % of LNCaP DHRS7 levels observed in PC3 cells calculated from normalizing to the tubulin signal is given underneath the lanes. (**C**) Antibody staining for DHRS7 in LNCaP cells reveals a strong nuclear envelope accumulation along with diffuse staining in the ER. Notably, levels are very low in the bottom cell indicating expression variability in the population. Scale bar, 10 µm. (**D**) A model for DHRS7 transmembrane helices was generated using TMHMM v2.0. (**E**) Determination of topology using the digitonin assay. Cells expressing a DHRS7-mRFP fusion (red in the merged image) were permeabilized with either Triton X-100 which removes all membranes or digitonin which preferentially pokes holes in the plasma membrane where cholesterol is abundant. Cells were then fixed and stained with an antibody against mRFP (Alexa-488, green in the merged image). If the epitope is in the lumen it is masked and so will be inaccessible to the antibody so that there would be less signal from the antibody than from the mRFP. The similar staining pattern indicates the epitope on the C-terminal region of the protein is accessible to the cytoplasm/nucleoplasm. Scale bar 10 μm.

**Figure 2 cells-13-00057-f002:**
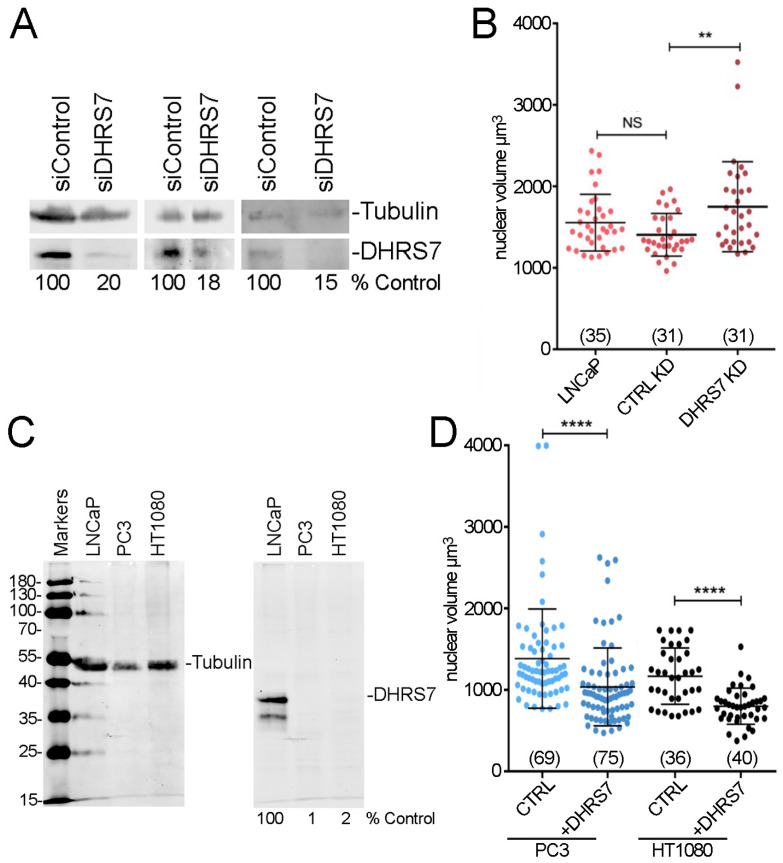
DHRS7 can direct nuclear size changes in prostate cancer cells. (**A**) Protein levels of DHRS7 by Western before and after siRNA knockdown in LNCaP cells from 3 biological replicates. The % of DHRS7 signal compared to the siRNA control after tubulin normalization is given under the lanes. (**B**) Whisker-scatterplot chart showing quantification of volume reconstructions for LNCaP cells knocked down for DHRS7 shows its loss resulted in increased nuclear size. (**C**) Western blot as in Figure 1 showing also levels in HT1080 cells. (**D**) Whisker-scatterplot chart showing the quantification of volume reconstructions for PC3 and HT1080 cells with or without exogeneous DHRS7 expression as indicated. Exogenous expression of DHRS7 in late-stage PCa PC3 cells yields decreased nuclear size. For (**B**,**D**) the number of cells analyzed is shown in parentheses at the bottom. All statistical analyses were performed with unpaired *t*-test with Welch’s correction: ** *p* < 0.01, **** *p* < 0.0001.

**Figure 3 cells-13-00057-f003:**
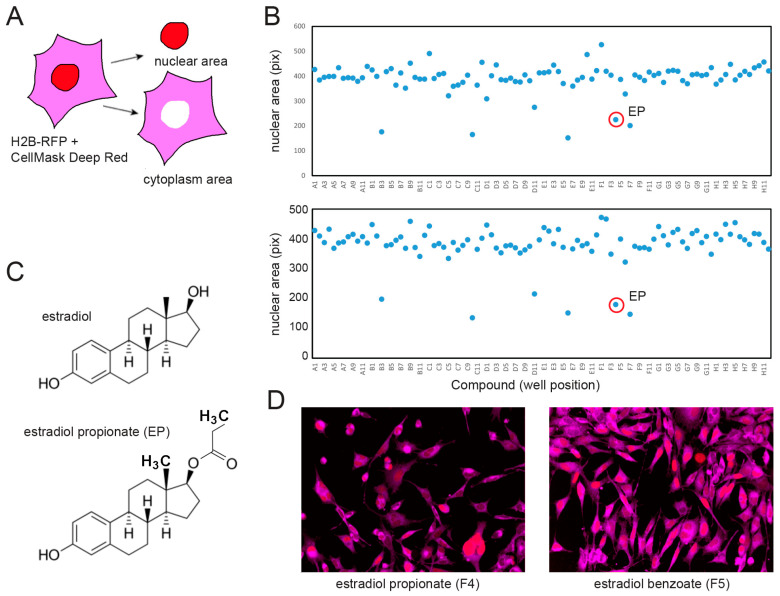
Medium-throughput screen for compounds decreasing nuclear size in PC3 cells. (**A**) Screening approach. Drug-treated PC3 cells stably expressing H2B-RFP to determine nuclear size were stained after fixation with CellMask Deep Red dye to determine the total cell area. (**B**) Replicate screens of plate 6 from the Microsource Spectrum library (compounds listed on Appendix A, concentration: 10 μM). Nuclear size (vertical axis) was averaged over a total of 350–4000 cells per well (horizontal axis). A1/A12, B1/B12 … G1/G12: DMSO controls. Estradiol propionate (EP) is indicated (red circles). Nuclear size reduction by at least 20% was statistically significant (*p* < 0.001, Wilcoxson rank test). (**C**) Molecular structures of estradiol (top) and EP (bottom). (**D**) Characteristic images of EP (**left**) and estradiol benzoate (**right**)-treated cells (blue: H2B-RFP nuclear signal; red: CellMask Deep Red cytosolic signal).

**Figure 4 cells-13-00057-f004:**
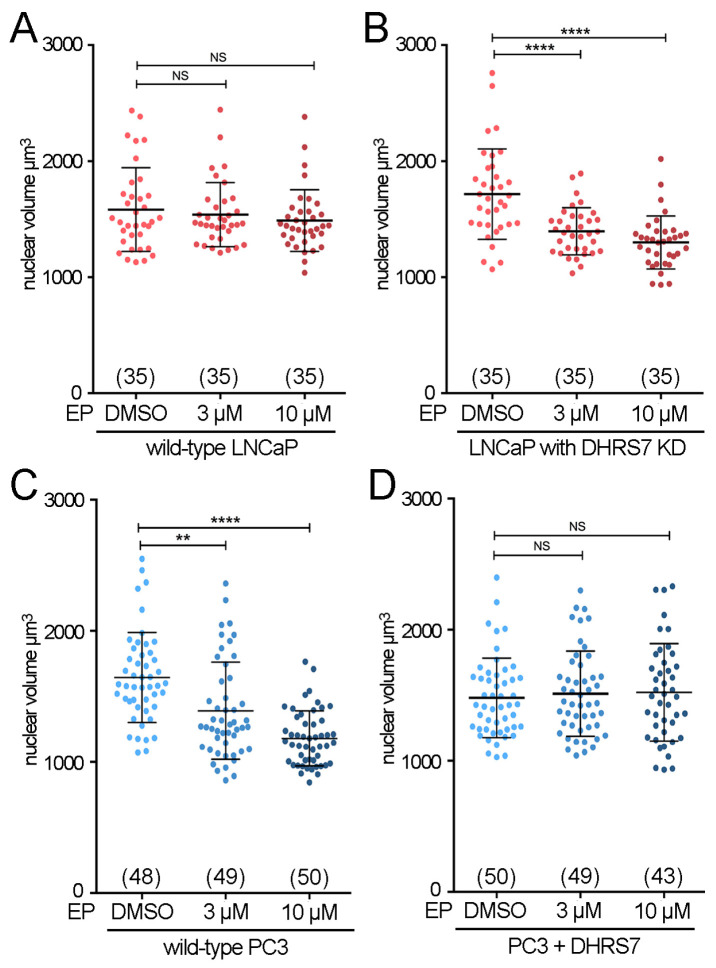
Estradiol propionate only alters nuclear size with low DHRS7. (**A**–**D**) Whisker-scatterplot charts showing the quantification of nuclear volume in cells in presence of the indicated concentrations of estradiol propionate (EP). (**A**) Wild-type LNCaP cells. (**B**) LNCaP cells with DHRS7 knockdown. (**C**) Wild-type PC3 cells. (**D**) PC3 cells with exogenously restored DHRS7 expression. For all conditions, the number of cells analyzed is shown in parentheses at the bottom of the graph. All statistical analyses were performed with unpaired *t*-test with Welch’s correction: ** *p* < 0.01, **** *p* < 0.0001.

**Figure 5 cells-13-00057-f005:**
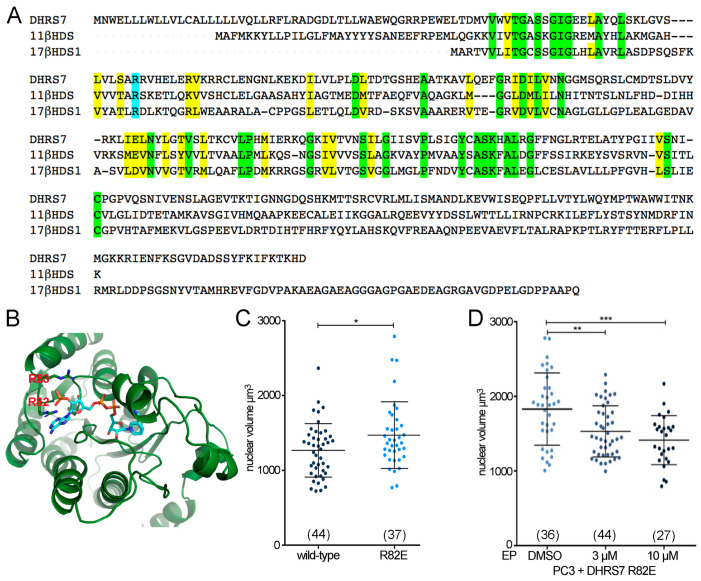
DHRS7 dehydrogenase activity is likely required for its effects on nuclear size. (**A**) Sequence alignment of DHRS7 against 11β-HDS-B1 and 17-β-HDS-1. The mutated arginine (R82 in DHRS7) is highlighted in blue while residues conserved are in green and conservation of amino acid functionality is indicated in yellow. (**B**) Structural prediction for DHRS7 generated in April 2017 with the Phyre2 web server, www.sbg.bio.ic.ac.uk/phyre2/, using human 11β-HDS-B1—PDB 2ILT 1BHS. (**C**,**D**) Whisker-scatterplot charts showing the quantification of nuclear volume in PC3 cells expressing DHRS7 wild-type or R82E mutant alone (**C**) or in presence of the indicated concentrations of estradiol propionate (EP) (**D**). For (**C**,**D**) the number of cells analyzed is shown in parentheses at the bottom of the graph for each condition. All statistical analyses were performed using unpaired *t*-test with Welch’s correction: * *p* < 0.05, ** *p* < 0.01, *** *p* < 0.001.

## Data Availability

Images from the OPERA screening platform are stored in an archive datastore at the University of Edinburgh and could be made available upon request. Images used for quantification in directed experiments are available on the university COIL server and again can be retrieved and made available upon request.

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
