# Peer review of "Reduction in Nuclear Size by DHRS7 in Prostate Cancer Cells and by Estradiol Propionate in DHRS7-Depleted Cells"

_cells, 2023, doi:10.3390/cells13010057_

Round 1

Reviewer 1 Report

Comments and Suggestions for Authors

The manuscript entitled 'Estradiol Propionate Reduction of Nuclear Size in Prostate Cancer Lines Lacking DHRS7 Suggests DHRS7 Absence as a Biomarker for the Effectiveness of Estrogen Therapy' investigates the impact of altered DHRS7 expression on nuclear size within prostate cancer cell lines. The study is well-articulated, utilizing robust scientific language and presenting a sound experimental design. Results are clearly delineated and appropriately analyzed. With minor revisions, this manuscript can be accepted for publication.

Questions:

1. In the manuscript, it is noted that the PC3 and HT1080 cell lines were cultured in DMEM. However, according to the American Type Culture Collection (ATCC) guidelines, the recommended growth media for PC3 is F-12K, and for HT1080, it is EMEM. Could the authors explain their departure from the recommended media for these cell lines?

2. Could the authors specify the duration of the blasticidin selection process applied to the cells?

3. The Western Blot images presented appear to lack numerical normalization. Authors should consider providing normalized data.

4. In the description of the figures, it appears that the sample sizes (n) are not mentioned. Could the authors provide this information to clarify the number of observations or experiments that the figures are based on?

Author Response

The manuscript entitled 'Estradiol Propionate Reduction of Nuclear Size in Prostate Cancer Lines Lacking DHRS7 Suggests DHRS7 Absence as a Biomarker for the Effectiveness of Estrogen Therapy' investigates the impact of altered DHRS7 expression on nuclear size within prostate cancer cell lines. The study is well-articulated, utilizing robust scientific language and presenting a sound experimental design. Results are clearly delineated and appropriately analyzed. With minor revisions, this manuscript can be accepted for publication.

We thank the reviewer for their positive comments.

  1. In the manuscript, it is noted that the PC3 and HT1080 cell lines were cultured in DMEM. However, according to the American Type Culture Collection (ATCC) guidelines, the recommended growth media for PC3 is F-12K, and for HT1080, it is EMEM. Could the authors explain their departure from the recommended media for these cell lines?

First, re the historical reason for our departure from ATCC recommendations, we used recommendations from other local labs using these cell lines. My lab generally first does pilot experiments using cells from local colleagues and then, if we pursue a particular project, we purchase fresh cells from ATCC to ensure that we are as close to the original characterized cells as possible. Thus, we used the other lab's recommendations during our pilot experiments and continued using these media when we got fresh cells from ATCC.

Though many ATCC guidelines are historical, often from before other media formulations were available, we were concerned that we might have been in error and so sought to further investigate the different media. As far as the differences in the media themselves, it is worth noting that DMEM and Eagles MEM are reasonably similar with DMEM having everything at the same level or more as in Eagles MEM with the exception that MEM has more arginine, a bit more NaCl, and a bit more NaPhosphate.  DMEM also has additional amino acids and a salt that are absent in MEM. Thus, the EMEM+NEAA is even more similar to DMEM.  The difference between DMEM and F12-K medium is greater with DMEM having much higher concentrations of several amino acids but F12-K having 5 additional amino acids not in DMEM. F12-K also has a couple extra salt ions (Cu and Zn), two extra vitamins (Biotin and B12), and additional components like hypoxanthine, linoleic acid, lipoic acid, putrescine, Na pyruvate, and thymidine.  However, the reason stated for this formulation was because it was often used with lower serum levels and these components were to make up for the absence of serum.  Since we are using 10% serum, many such components are nonetheless present.

Finally, regarding justification for our choice of media in the scientific literature, we performed a PubMed search for "PC3 cells and prostate cancer" to see what others are doing. The first 10 papers that came up (PMIDs 30719778, 31718684, 31752117, 31874106, 36591803, 30166521, 35981451, 37299000, 37273236, 31361600) had only 1 using the ATCC-recommended F12-K media for PC3 cells, while 4 used the same DMEM formulation we used, and 5 used the RPMI formulation we used separately for the LNCaP cells.

  1. Could the authors specify the duration of the blasticidin selection process applied to the cells?

The cells were maintained in blasticidin during the whole selection process of roughly a week. Subsequently, cell passages were alternated between having no blasticidin present or having the same 8 µg/ml blasticidin present in order to maintain selection.  All experiments were performed in the absence of blasticidin to ensure it would not potentially interfere with results and because only cells maintaining the expression could be visualized for quantification during the OPERA screen.  We have made this more clear in the Methods section, changing the sentence from:

"To generate PC3-H2B-mRFP, H2B-mRFP plasmid carrying the blasticidin resistance gene was linearized and co-transfected with plasmid pToI2 encoding for transposase using JetPrime (PolyPlus transfection) transfection reagent. 24 h post transfection the medium was replaced with medium containing 8 μg/ml blasticidin (Thermo Fisher Scientific) to select stably-integrated transfectants. After amplification the cells were FACS sorted for the brighter mRFP population that retained normal morphology."

to

"To generate PC3-H2B-mRFP, H2B-mRFP plasmid carrying the blasticidin resistance gene was linearized and co-transfected with plasmid pToI2 encoding for transposase using JetPrime (PolyPlus transfection) transfection reagent. 24 h post transfection the medium was replaced with medium containing 8 μg/ml blasticidin (Thermo Fisher Scientific), which was maintained for ~1 week to select stably-integrated transfectants. After amplification the cells were FACS sorted for the brighter mRFP population that retained normal morphology. Subsequently, passages of cells were alternated between selection and no selection; however, cells used in experiments were always following a passage without blasticidin and the screening was performed in the absence of blasticidin."

  1. The Western Blot images presented appear to lack numerical normalization. Authors should consider providing normalized data.

We have repeated the Western blots as we could not find the original LiCOR files and quantified them for the revised version.  For this we used both samples the student still had in the freezer for Figure 1B and since he had not tested the HT1080 cells we also freshly thawed all three cell lines and did a new quantification on these for Figure 2C. To ensure that knockdowns under the student condition are reproducible, we show in the revised Figure 2A three biological replicates from the student and the newly thawed cells in the revision period. In all cases, the new gels were quantified and normalized to the tubulin staining and we have added numbers for the percent of control to the bottom of the figure panels under the lanes. The method of quantification has also been added to the Methods section. "Scans were opened in Fiji and standard measurements taken for identically-sized region of interests of bands and background. Median values were extracted and background from the same lane subtracted. Corrected DHRS7 values were normalized to tubulin loading and percentage of the LNCaP control DHRS7 signal calculated for other lanes."

  1. In the description of the figures, it appears that the sample sizes (n) are not mentioned. Could the authors provide this information to clarify the number of observations or experiments that the figures are based on?

Thank you for raising this point.  While we typically use 100 cells for quantifying area with 2D analyses, all of the analysis shown here is 3D which takes a lot longer; so the student would often determine optimal sample sizes by performing statistical analysis as the data was being generated. As a result, the numbers are rather variable.  The student has sent me the numbers and we have added them in the figures themselves on the graphs in parentheses under each condition and accordingly noted this in the figure legends.

Reviewer 2 Report

Comments and Suggestions for Authors

The authors investigated the DHRS7 gene in prostate cancer cell lines by over-expression and knockdown studies. Knockdown of DHRS7 in LNCaP increased nuclear size while nuclear size was decreased with overexpression of DHRS7 in PC3 cells. Treatment of PC3 cells lacking DHRS7 with estradiol propionate also reduced nuclear size, but the effect of estradiol propionate was lost in presence of DHRS7. Mutational analysis indicated that the nuclear size-regulatory function of DHRS7 is dependent on its enzymatic activity.

The authors provide interesting preliminary data, but their data do not support their conclusions. Only one cell line is used for each condition. The authors should perform DHRS7 knockdown experiments in at least one other (androgen-sensitive) cell line and DHRS7 over-expression in at least one other (androgen-insensitive) cell line. HT1080 cells are a fibrosarcoma cell line and are not derived from prostate cancer. There is no data presented addressing whether DHRS7 absence is a biomarker for the effectiveness of estrogen therapy.

Other points:

Figure 2: western blot is missing for PC3 and HT1080 cells.

Figure 3: panels C and D are incorrectly labelled.

Figure 4: experiments should be performed with additional cell lines

Comments on the Quality of English Language

No issues with quality of english.

Author Response

The authors investigated the DHRS7 gene in prostate cancer cell lines by over-expression and knockdown studies. Knockdown of DHRS7 in LNCaP increased nuclear size while nuclear size was decreased with overexpression of DHRS7 in PC3 cells. Treatment of PC3 cells lacking DHRS7 with estradiol propionate also reduced nuclear size, but the effect of estradiol propionate was lost in presence of DHRS7. Mutational analysis indicated that the nuclear size-regulatory function of DHRS7 is dependent on its enzymatic activity.

The authors provide interesting preliminary data, but their data do not support their conclusions. Only one cell line is used for each condition. The authors should perform DHRS7 knockdown experiments in at least one other (androgen-sensitive) cell line and DHRS7 over-expression in at least one other (androgen-insensitive) cell line. HT1080 cells are a fibrosarcoma cell line and are not derived from prostate cancer. There is no data presented addressing whether DHRS7 absence is a biomarker for the effectiveness of estrogen therapy.

The experiment that would have been ideal to strengthen this manuscript would be to actually test what we are proposing as a follow up experiment, i.e. checking the levels of DHRS7 in tumors from patients treated with estrogen therapy and correlate this with their response to the therapy. However, it is nearly a decade since the last study we could find published was concluded and we were not able to identify clinicians who had remaining patient samples before and after treatment to give us to test. Yet, we think it is very important that this additional direction/hypothesis suggested by our findings gets presented so that clinicians using estrogen therapy (which has gone in and out of popularity several times since the 1970s) in the future can check it when samples become available. At the same time, we appreciate the reviewer's comments as we very much would have liked to have this data too. Therefore, to address their comments in this regard, we have made slight modifications throughout the text to emphasize that at this point it is just a suggestion that DHRS7 might be a good biomarker for the effectiveness of estrogen therapies. Most notably, we have changed the title from "Estradiol Propionate Reduction of Nuclear Size in Prostate Cancer Lines Lacking DHRS7, Suggests DHRS7 Absence as a Biomarker for the Effectiveness of Estrogen Therapy" to "Estradiol Propionate Reduces Nuclear Size in Late-Stage PC3 Prostate Cancer Cells With Reduced DHRS7, But Not in Early-Stage LNCaP Cells or PC3 Cells Expressing DHRS7".

Other points:

Figure 2: western blot is missing for PC3 and HT1080 cells.

This was definitely an oversight on our part.  The student evidently was simply relying upon BioGPS indications of background DHRS7 transcript levels in HT1080 cells and so never ran a Western on this.  Accordingly, we have thawed all three cell lines and done new Western blots to compare them, now added as Figure 2C.  As expected, we see no signal for the HT1080 cells or, again, for the PC3 cells. The point of testing the HT1080 cells, to address the general comments above, was to see if exogenous expression in a completely different cell line and cancer type lacking DHRS7 could also increase nuclear size. We have made slight modifications to the text to make this more clear.  

Figure 3: panels C and D are incorrectly labelled.

Thank you for pointing this out and our apologies for the mistake. This has now been corrected.

Figure 4: experiments should be performed with additional cell lines

Regarding the specific experiment suggested, we agree that it would be nice to show the same effects and principles apply when comparing another early-stage PCa model with another late-stage androgen-insensitive model. Accordingly, we asked if any colleagues had DU-145 cells to, like PC3, reflect late-stage androgen-insensitive tumors and VCaP cells to like LNCaP reflect an early androgen-sensitive stage (note that several of the other commonly used early stage models such as C4-2 and C4-2B lines are actually derived from LNCaP cells passed through mice in xenograft models, so are essentially LNCaP); however, we could not get a complete set to compare with the LNCaP and PC3 cells we used. Without having a complete set this experiment would be largely meaningless and based on previous experience it would take at least a month and a half to get the new cell lines from ATCC and do the experiments when we were given just 10 days for revisions.  While there are some other PCa lines available, they are mostly later-stage androgen sensitive, so that they would be asking a different question. While it would also be also nice to compare these against androgen-insensitive earlier stage and castration-resistant lines that nonetheless still express androgen receptor such as 22Rv1and ARCaP lines, this begins to be a new study asking additional questions and so was not pursued.

We would further note that recent publications on the cell biology of prostate cancer indicate that results obtained with 2 cell lines are well within current standards, including for publication in Cells.  We performed a PubMed search for "PC3 cells and prostate cancer" to see what others are doing. The first 10 papers that came up (PMIDs 30719778, 31718684, 31752117, 31874106, 36591803, 30166521, 35981451, 37299000, 37273236, 31361600) had 3 papers using only PC3 and not testing any other PCa lines, 2 papers which like us used 2 PCa cell lines, 1 paper using 3 PCa lines, and 4 papers using 4 PCa lines.  Notably, one of the three papers using just PC3 cells was published in the MDPI journal Moleculeswhile one of the papers using just two PCa lines was published in Cells. Thus, we are consistent with the standards in the field. 

Round 2

Reviewer 2 Report

Comments and Suggestions for Authors

There still may be an issue with labeling for Fig 3C and Fig 3D. Otherwise the manuscript is now acceptable.

Author Response

Response: We are pleased that the reviewer has stated that the manuscript is now acceptable for publication.  With respect to the comment about Figure 3 labelling, I had 3 authors separately check the revision Figure 3 in response to this comment and all concurred that the C and D labels in the figure match the text in the Results section and Figure Legend and moreover that the structures for estradiol and estradiol propionate are correct.  Thus, I am particularly grateful that the reviewer's comment inspired me to take a closer look, upon which I noticed that in the legend for panel D it stated that the H2B-RFP signal was blue in the images and the cell mask in red when in fact the H2B-RFP signal is red and the cell mask signal is magenta.  This was an oversight as in earlier versions of the figure we had false-colored the nucleus blue since that is what people are used to seeing, but then we noted that since the screen as described in panel A was using the H2B-RFP to delineate the nucleus that it might make more sense to show the images the same way to avoid confusion; however, we never changed this in the legend.  Thus, we again thank the reviewer for noting that something was off and we have now corrected the figure legend to match the images.

Note that in this careful review of the figure and text we also thought that the term "highlight" would be more appropriate than "indicate" for the red circles in panel B and so also changed this in the figure legend.  Moreover, we thought the text might have been slightly confusing where we point out the two other estradiol derivatives and then refer to panel D where we only show images of one.  Therefore, we have also changed this text to refer to panel B for both compounds and then more clearly explain that we only show images from one in panel D.